# Antecedents of Rural Tourism Experience Memory: Tourists’ Perceptions of Tourism Supply and Positive Emotions

**DOI:** 10.3390/bs12120475

**Published:** 2022-11-24

**Authors:** Hang Chen, Yuewei Wang, Minglu Zou, Jiaxin Li

**Affiliations:** 1School of Tourism Management, Shenyang Normal University, Shenyang 110034, China; 2School of Business, Liaoning University, Shenyang 110036, China

**Keywords:** rural tourism, perceptions, positive emotions, tourism experience memory

## Abstract

Tourism experiences bring about physical or psychological feelings in tourists, which can not only leave tourists with deep memories, but also affect their behavioral intentions. Tourism experiences are meaningful only if they can be remembered and influence word of mouth and decision making. A better understanding of what influences tourism experience memory will help optimize the supply and further development of tourism destinations. This study explores the antecedents of rural tourism experience memory from the tourism supply perspective, revealing the mechanism of effect of these antecedents on tourists’ tourism experience memory formation through a questionnaire-based survey of 556 participants in Xidi Village, China, and correlation and multiple regression analyses. The results show that perceptions about the supply of rural tourism destinations trigger positive emotions that, in turn, affect the formation of rural tourism experience memory. Through the mediating role of positive emotions, there is a significant correlation between perception of rural tourism destination supply and the formation of rural tourism experience memories.

## 1. Introduction

In this experience economy era, rural tourists are no longer satisfied with agricultural sightseeing tours but prefer immersive tourism experiences. While traveling, tourists seek spiritual satisfaction, unforgettable impressions, and memorable tourism experiences. Such experiences are preserved as memories in the minds of tourists, and those memories will have an influence on tourists’ future decisions [1]. These memories reflect tourists’ perceptions, cognitions, and emotions [2]. They may also reflect anything that a tourist destination offers tourists. Thus, today, experiential memories are key to influencing tourists’ decisions about rural tourism destinations. Therefore, a better understanding of what influences tourism experience memory will help optimize the supply and further development of tourism destinations. Such an understanding can be reached by generating and reflecting on tourism experience memory.

Research on the antecedent variables of tourism experience memory, which mainly considers two aspects, has attracted the attention of scholars. One aspect is exploration based on the perception of visitors. For example, Kim put forward seven antecedent variables of memorable tourism experiences: hedonistic, recovery, involvement, novelty, meaning, knowledge, and local culture; this proposal is the most widely accepted suggestion at present [3]. Mahdzar et al. confirmed the effect of perceived quality on memorable tourism experiences in an empirical study of Mulu National Park [4]. Therefore, rich perceptual elements are conducive to strengthening tourists’ memory of travel experience. Secondly, Kim et al. argued that tourists’ perception factors are not enough to reflect the whole problem, demonstrating that attributes surrounding the destination such as the local culture, activities and special events, destination residents’ attitudes, infrastructure, accessibility, environmental management, service quality, natural tourism resources, cultural tourism resources, and local attachment to the destinations’ attributes affect memorable tourism experiences [5]. However, not all perceptions about rural tourism destination supply can form unforgettable tourism experience memories. Generally speaking, tourism experiences that elicit a strong emotional response are the ones that are memorable. Perceptions about rural tourism destination supply incite emotions, which permeate the experiences of tourists and affect their attitudes, memories, behaviors, and other reactions. Therefore, this study regards the positive emotions generated by perceptions about rural tourism supply as important antecedent variables for the formation of rural tourism experience memory, and it is very possible that positive emotions play a mediating role between rural tourism supply perception and tourism experience memory. In the context of China’s rural tourism environment, this study attempts to analyze the antecedent variables of tourism experience memory and explore how supply perceptions and positive emotions related to rural tourism destinations are transformed into tourism experience memory. The study also clarifies whether positive emotions play a mediating role between perception about rural tourism destination supply and tourism experience memory. Rural tourism managers realize the importance of tourism experience memory and attempt to explore the formation mechanisms of rural tourism experience memory; that is, the supply perception–positive emotions–tourism experience memory structure could help these managers formulate targeted marketing strategies, product designs, and promotions to provide tourists with unforgettable memories and boost local tourism.

This paper attempts to explore the antecedents of rural tourism experience memory in the context of tourists in rural China, and reveal the mechanism of the effects of perception about rural tourism destination supply and positive emotions, on tourism experience memory.

## 2. Literature Review and Hypotheses Development

### 2.1. Tourism Experience Memory

Memory is the accumulation of people’s impressions of activities and behaviors that have experienced in their minds [6]. In the case of tourism experiences, both the expected and on-site experiences are stored in the mind of the tourist in the form of memories, referred to here as tourist experience memory, which may gradually disappear spontaneously or due to interference factors, or may be strengthened continuously by stimulus factors. When tourists are faced with a travel decision again, their stored experience memories from past travels will be extracted in an orderly manner, causing them to recall the unique feelings experienced previously. Here, they may rely on those experience memories to make their destination decision [7,8]. Therefore, only when the tourist experience is transformed into a tourism experience memory that is deeply remembered by the tourist, can the tourist’s behavioral intentions, such as revisiting the destination or recommending it to someone else, be affected [9]. Tourism experience is the accumulation of individual tourists’ experiences—based on emotions and occurrences—during their tours. Not all tourism experiences will be vividly and permanently remembered by tourists [10]. Kim pointed out that memorable tourism experiences are more likely to stimulate the formation of flashbulb memories of tourists, which are characterized by brightness, accuracy, and persistence [11]. Based on the results of Kim et al.’s research [12], this study defines the vivid long-term memories formed by the memorable tourism experiences of tourists as tourism experience memory. The two most important dimensions of tourism experience memory are reproducibility and vividness of memory [13]. Reproducibility refers to the complete recall of the tourism experience, while vividness refers to the recall of specific things, plots, and emotions in the tourism experience.

### 2.2. Tourists’ Perceptions of Destination Supply

Scholars of tourism usually classify dimensions of destination supply into three tourism products, namely, attractions, services/facilities, and transportation [14,15,16,17]. However, diversified promotion and information are also important ways for tourists to form memories of their tourism experiences [18]. Holbrook et al. believe that the essence of experience is pleasure, which is realized through sensory perception [19]. Lemon et al. further proposed that the customer’s perception of the company’s products during the whole purchasing process constitutes a part of the experience [20]. Becker et al. believe that customer experience is an “unintentional and spontaneous response” to a specific stimulus [21]. In this study, the dimensions of perception about rural tourism destination supply are divided into four aspects: tourism attraction perception, service/facility perception, information perception, and promotion perception, which consider not only tourists’ on-site experiences, but also tourists’ pre-experiences.

### 2.3. Positive Emotions

Most studies show that emotions are positive or negative experiences associated with specific patterns of physical activity [22,23,24,25]. Positive emotions are defined as positive psychological states such as joy, surprise, happiness, and pride that tourists feel. Since the pathway of influence of positive and negative emotions is relatively independent, we can focus on the role of positive or negative emotions separately according to research needs [26]. Hosany et al. stated that a single-level dimension is more effective than a two-level dimension in measuring tourists’ emotional experiences, while the positive emotion dimension can better reflect the value of tourism [27]. First, tourism is a series of events in which tourists seek pleasant and unforgettable experiences for enjoyment. The “rosy view” effect may reduce negative phenomena in tourists’ reviewing of a tourism experience and magnify positive experiences [23]. Second, negative memories are characterized by defense and avoidance, and tourists tend to seek advantages and avoid disadvantages by forgetting unpleasant negative events and reconstructing the overall experience to reduce cognitive dissonance [15]. Therefore, this study focuses on the positive factors in tourist experiences and on the mechanism of positive emotions. Hosany et al. believed that emotion is the result of an individual’s processing or evaluation of relevant information, such as situations or events, and develops a concise destination emotion scale (DES) which mainly contains three dimensions of positive emotions: joy, love, and positive surprise [27,28]. In fact, positive emotions arising from tourists’ perceptions of destination supply can contribute to an individual’s overall tourism experiences by enhancing their sense of well-being [29,30].

### 2.4. Hypotheses Development

According to the emotional evaluation theory, emotions are generated through the individual’s internal perception evaluation mechanism, and tourists’ perception evaluation of the destination supply determines their emotional response [31]. In addition, Hosany and Breitsohl, when exploring the influencing factors of tourists’ emotions, believed that the perception of destination supply was one of the important sources of stimulation affecting tourists’ emotional state [32,33]. In the face of a good tourism environment, resources, facilities, services, and other destination supply factors, tourists usually have positive emotional experiences such as surprise and joy. To reduce purchase risks, tourists tend to collect relevant product information and promotion schemes before making decisions to satisfy their imaginations and assumptions about various aspects of the destination, such as scenery and service facilities, and thus trigger the emotional reaction of tourists expecting to participate in local tourism activities. Lamore et al. believed that the integration of products and promotional activities produces a synergistic effect, which helps consumers generate positive emotions [34]. Long verified that destination information supply, perceived promotion, and positive emotions, such as happiness, gratitude, and passion, had a significant impact on tourists by applying emotion evaluation theory and the hierarchical regression method [35]. Tercia believed that the contact of stimuli such as information promotion at all stages of tourism causes tourists to react emotionally [36]. Therefore, we propose the following hypotheses:

**Hypothesis** **1.**
*The four dimensions of tourism supply perceptions, that is, tourist attraction perception, service/facility perception, promotion perception, and information perception, included in destination supply perception have a significant positive impact on the three dimensions of tourists’ positive emotion composition, namely, joy, love, and positive surprise (divided into 12 subdivided hypotheses).*


In Bohanek’s (2005) study, emotions had a significant impact on the depth of memories, and people were more likely to recall experiences that involved many emotions and recalled them more vividly [6]. Events related to positive emotions more easily form deep experiential memories compared to those without emotional involvement [11], and the memory recall process is faster and more accurate. Tung believes that positive emotions are one of the most important components in promoting the formation of tourists’ (positive) experience memories [9]. Kim believes that positive emotions have a profound influence on the vividness and reproducibility of tourists’ experience memories [5]. Pan conducted a study on the Chinese situation based on Kim’s study and found that tourists’ positive emotions had a significant positive influence on the reproducibility and vividness of their experience memory [13]. Through interviews, she found that tourists are often better at remembering pleasant experience memories, are willing to record and analyze them, and hope to get positive affirmation of the travel experience in the process of sharing [37]. Curiosity is an important internal motive driving individual behavior and seeking surprise is an important motive for tourists to travel [7,38]. Tourists tend to look for places with unique tourism resources. After the tour, the first memory of the trip is often the most surprising experience. Therefore, this study proposes the following hypothesis:

**Hypothesis** **2.**
*The three components of tourists’ positive emotions, including joy, love, and positive surprise, have a significant positive impact on the two components of vividness and reproducibility of tourism experience memory (six subdivided hypotheses).*


The supply of tourist destinations is the basis of tourist experience, and the memories formed through tourist experience are the basis of the supply of the destination. Barbieri et al. studied surfing tourism and revealed that tourists’ perceptions of perfect waves, exotic natural environments, and other perceptions of destination supply influence tourism experience memory [39]. Kim constructed a destination attribute measurement scale related to tourism experience memory, laying the foundation for conducting further research on the creation of tourism experience memory [5]. Lee analyzed the role and influence of experiential memory in a restaurant reconstructed from an old railway station in Taiwan, indicating that there exists a strong correlation between tourists’ perceptions of the cultural heritage of food attractions and feelings of nostalgia and experiential memory [23]. Pan et al. found that tourists’ perceptions of the unique supply features of certain tourist destinations significantly promoted the vividness of tourists’ experience memory [13]. Wannoo believed that pricing policy, uncrowded slopes/lifts, beautiful scenery, and accessibility had a significantly positive impact on the vividness, coherence, accessibility, and other characteristics of experience memory [40].

In addition to the influence of the above-mentioned tourism product perceptions on tourism experience memory, some scholars have proposed that information and promotion programs for tourist destinations may also have an important impact on tourism experience memory. For example, Simon took Canadian brand promotion activities as an example and proposed that tourists’ perceptions of destination information and promotion had a significant impact on tourism experience memory [41]. Kruger et al. found that tourism experience memory was affected by the quality of information about food and entertainment, and of promotional material [42]. Papyrina proposed that advertising information is used as an external stimulus to resonate with consumers’ imagination, arouse tourists’ emotions, and convey specific memories of the pre-visit experience [43]. Therefore, this study proposes the following hypothesis:

**Hypothesis** **3.**
*The four component dimensions of the perception of destination supply, including tourist attraction perception, service/facilities perception, promotion perception, and information perception, have a significant positive impact on the two dimensions of tourism experience memory: vividness and reproducibility (eight subdivided hypotheses).*


As discussed above, tourists’ perceptions of destination supply have been found to positively influence tourism experience memory. Furthermore, the literature suggests that tourists’ perceptions of destination supply and positive emotions are positively correlated, which affects tourism experience memory. Accordingly, this study examines positive emotions as a mediator between tourists’ perceptions of destination supply (specifically, tourism attraction perception, service/facilities perception, promotion perception, and information perception) and tourism experience memory in rural tourism. That is, the study explores tourism experience as an external stimulus of tourists’ positive emotions which, in turn, may influence the formation of more profound and unforgettable experience memories. This is because the internal influence mechanism of destination supply perception on tourism experience memory may be realized through the mediating role of positive emotions (Figure 1). Accordingly, this study proposes the following hypothesis.

**Hypothesis** **4.**
*Positive emotions*
*mediate the relationship between tourists’ perceptions of destination supply and tourism experience memory.*


## 3. Methodology

A questionnaire survey was used in this study. The target participants included tourists to the ancient villages of Xidi in Anhui Province, China. The questionnaire was designed using the concepts developed based on the studies explored in the literature review. The questionnaire was pre-tested among 100 visitors and the modifications were made based on the results of that pilot study.

### 3.1. Instrumentation

Demographic variables such as age, gender, and marital status were investigated to determine the explanatory variables and to compare the results with those of other studies. The selection of the travel characteristic variables was conducted with reference to other rural tourism related studies [44,45,46], including duration of stay and travel mode.

The measurement of tourists’ perceptions of destination supply was based on Buhalis, Sun, and Gunn, and from interviews with ten tourists [16,18,47,48]. For tourist attraction perception, the survey was conducted from three perspectives of natural landscape, spatial pattern design, culture, and art. At the level of service/facility perception, the survey was conducted from four perspectives: service quality, service attitude, service facilities, and communal facilities. For promotion perception, the survey was conducted from two perspectives: promotional activity and promotion scheme. For information perception, the survey was conducted from three perspectives: inquiry channels, intelligent tourism information services, and intensive advertising. To measure these aspects, a 7- point Likert scale was adopted, where the higher the score, the better the tourists’ perceptions of destinations supply.

Furthermore, participants in this study completed a self-reported 12-item questionnaire developed by Kim, She, and Hosany et al. [11,27,37]. To evaluate “joy”, we measured three aspects: carefreeness, delight, and fun. For “love”, we measured four aspects: friendliness, enthusiasm, gratitude, and feeling at ease. For positive surprise, we measured four aspects: novelty, shock, revel, and meaning. To measure these aspects, a 7- point Likert scale was adopted, where the higher the score, the more intense the emotions of the interviewee.

Tourism experience memory was measured according to Kim, Chandralal, and Pan et al. [11,13,49]. To evaluate vividness, the tourists’ memories of the layout and structure of the main scenic spots, the sounds they heard, and the scenes they saw were investigated from four perspectives: overall process, the layout and structure of the main scenic spots, the sounds they heard, and the scenes they saw. For reproducibility, we measured four aspects: reproducing the tourist experience recall, participating in the activity recall, emotional feeling recall, and major scenic spots recall. To measure these aspects, a 7-point Likert scale was adopted, where the higher the score, the deeper the memory of the interviewees.

### 3.2. The Study Setting

This survey was conducted in Xidi Village, which is located 40 km away from Mount Huangshan and 8 km east of Yi County. Xidi, a scenic Chinese folk culture village in Anhui Province, is an ideal spot to see the typical old dwellings of southern Anhui Province, where residents live as they have for hundreds of years. In 2000, Xidi Village was listed as a UNESCO World Heritage Site. There are 124 well-preserved dwellings in the village and three temples built during the Ming and Qing Dynasties. There is a landmark in front of the village, a memorial archway built for Hu Wenguang, a magistrate of Xidi Village in the Ming Dynasty. The houses exhibit features such as fine wood and brick carvings, which reflect the long history of local culture and customs. This place is called “the Land of the Peach Blossom” or “Home in Wonderland” because of its perfection and harmony. It is also called “a treasure house of ancient resident architecture” for its well-preserved memorial archways and ancient buildings dating back to the Ming and Qing Dynasties.

Visitors can stroll through the old village and slowly discover its beauty. Traditional Hui-style architecture, white walls with gray tiles, prominent horse-head walls, and stone slab bridges constitute the village’s environment. The street pattern of Xidi is mainly an east–west main road, with two parallel streets on both sides. The main streets are joined together by many narrow lanes. Many villagers have preserved several artworks, and visitors can see woodcarvings, murals, and garden exhibitions displayed by local villagers. The houses in Xidi Village have the same antique furniture and painting layout as those in the Ming and Qing Dynasties (1368–1911 AD). These factors make Xidi Village a prominent rural tourism destination today.

### 3.3. Sample

The respondents were informed in advance that the information they provide would be anonymous, strictly confidential, and used only for academic purposes. An on-site investigation with a self-administered questionnaire was conducted in Xidi Village, China. The investigation was conducted over several weekdays and weekends, during March and April, when the rapeseed is in full bloom, to ensure that the sample of tourists obtained is more comprehensive.

### 3.4. Data Analysis

Data analysis was performed using the Social Sciences version 22.0 statistical software package. The applied statistical methods used in this study include descriptive statistical analysis, reliability and validity analysis, principal component analysis, and correlation analysis. AMOS 22.0 was used in this study to conduct two-stage structural equation modeling (SEM) with the method proposed by Anderson and Gerbing [50]. Confirmatory factor analysis (CFA) was used to examine the psychometric characteristics of these scales. Subsequently, the general SEM was used to test the validity of the proposed model and hypotheses.

## 4. Empirical Analysis

### 4.1. The Respondents

The main survey was conducted between March and October 2019. A total of 900 questionnaires were distributed, and 756 valid questionnaires were finally returned. To determine the structure of each influencing factor, 200 responses were randomly selected and the remaining 556 responses were used to test the reliability and validity of the constructed model, as well as the hypothesis about the relationships between tourists’ perceptions of destination supply, positive emotions, and tourism experience memory. The demographic profile and travel characteristics of the survey respondents are as follows. The demographic profile and travel characteristics of the respondents are shown in Table 1. More than half of the respondents were single (67.7%) and a small number of respondents were married (32.3%). 339 (61%) respondents were male and 217 (39%) were female. Most of the respondents were aged between 16 and 25 years (62.2%), and the majority were students (51.6%). 377 (67.9%) respondents stayed for 1 to 1.5 days, 133 (23.8%) stayed for 1.5 to 2 days and 46 (8.3%) stayed for more than 2 days. 368 (66.1%) respondents arrived by tourist coach, 51 (9.1%) by bus, 127 (22.8%) by car and 10 (2%) by others. 49 (8.7%) respondents worked in the service sector, 29 (5.3%) in business, 24 (4.3%) housekeepers, 20 (3.7%) freelancers, 11 (2%) in other industries and 50 (8.9%) retired people. In addition, 315 (56.5%) respondents obtained information about rural tourist destinations from their relatives and friends, 73 (13.2%) based on personal experience, 117 (21.1%) from the Internet, 40 (7.1%) from Television/magazines, and 11 (2.1%) from travel agencies.

### 4.2. Reliability and Validity Analysis

Firstly, SPSS22.0 software was used in this study to test Cronbach’s α coefficient of the collected samples. The reliability coefficients of each component surface were mainly distributed between 0.716 and 0.807, all greater than 0.700, indicating good reliability (Table 2). The reliability standard is defined as the reliability coefficient greater than 0.5, which indicates that the samples collected in this survey have good reliability, and thus can be used for further analysis. Secondly, AMOS 22.0 software was used to test the fit degree of the model of the test factor analysis, and the relevant indicators of absolute fit degree and the relevant indicators of value-added fit degree reached the standard. Therefore, it was proved that the model constructed had a good fit degree with the data. This indicates that the three measurement scales of perception about tourism destination supply, positive emotion, and tourism experience memory all have good validity (Table 3, Table 4 and Table 5). Thirdly, in terms of acceptance validity of the validity test, the standardized load of each item was greater than 0.600 (between 0.662 and 0.861), reaching the applicable standard. The relative T values range from 11.827 to 36.755, which are also above the usable range of 0.600. The combined reliability CR ranged between 0.768 and 0.872, and the mean variance extraction AVE ranged between 0.524 and 0.641, both showing an ideal range of greater than 0.500. Therefore, the potential variables of the scale can be judged to have a good degree of convergence. In terms of the discriminant validity of the validity test, the root mean square of the AVE value of each latent variable was greater than its correlation coefficient with other latent variables, indicating that there is a significant difference among the variables. Based on this judgment, there is a good discriminative validity among the latent variables formed by the selected samples in the study. In conclusion, the samples have good reliability, the confirmatory factor analysis model has a good model fit degree, and the variables and their components have good validity. This indicates that the samples and scales have good reliability and validity and can thus be used for further hypothesis verification analysis.

### 4.3. Results of Hypothesis Testing

The verification results of Hypothesis 1 show that all the other 11 hypotheses subdivisions were verified, except that promotion perception had no significant effect on joy. The verification results of Hypothesis 2 show that “joy” has no significant effect on the vividness of rural tourism experience memory, and the test results of the other five hypotheses subdivisions are verified. The verification results of Hypothesis 3 show that perception about services/facilities has no significant impact on the vividness of rural tourism experience memory, and perception about promotional activity has no significant impact on the regeneration of tourism experience memory. The test structure of the other six hypotheses subdivisions has been verified Figure 2.

Referring to Bowen’s research results, the mediating effect test of the multiple regression analysis that was adopted here is as follows. Firstly, we analyzed the influence of the independent variable x on the dependent variable ($y=a_{0}+b_{0}x+\varepsilon_{0}$), and if the effect of the independent variable coefficient b_0_ was significant, we continued to the next step in the analysis. Secondly, we analyzed the influence of the independent variable x on the intermediary variable ($m=a_{1}+b_{1}x+\varepsilon_{1}$), and if the independent variable factor b_1_ was significant, we continued to the next step of the analysis. Finally, we inspected how the independent mediation variables affect the dependent variable ($y=a_{2}+b_{2}x+c_{2}m+\varepsilon_{2}$), and the influence of the intermediary variable on the dependent variable coefficient (c_2_) (Table 6). If the influence of the independent variable on the dependent variable is not significant (coefficient b_2_ is not significant), the mediator variable is a complete mediator variable. If the influence of the independent variable on the dependent variable is still significant (coefficient b_2_ is still significant), but the significance level is reduced, and the coefficient result of the regression of the independent variable to the dependent variable becomes smaller, then the mediating variable is a partial mediating variable. According to the above analysis, the regression analysis of tourism experience memory, taking destination supply perception as the independent variable and positive emotions as the dependent variables, revealed that the effect of other routes was significant, except the effect of promotion perception on joy and reproducibility. In addition, by applying demographic variables as control variables and positive emotions as the independent variables, the regression results showed that joy had no significant effect on vitality, but that the effects of other paths were significant. That is, further validation showed that positive emotions had a significant effect through its intermediary role path (the third mediation effect analysis). In Hypothesis 4, positive surprise played the strongest mediating role in the effect of rural tourist attraction perception on vividness, with the mediating effect of 0.098, accounting for 54.4% of the total effect. Love had the strongest mediating effect on reproducibility, with a mediating effect of 0.045, accounting for 36.9% of the total effect. Positive surprise had the strongest mediating effect on information perception, with a mediating effect of 0.063, accounting for 42.0% of the total effect, and also had the strongest mediating effect on the effect of promotion perception on vividness, with a mediating effect of 0.058, accounting for 26.8% of the total effect.

## 5. Discussion

Rural tourism experience memory is more and more important for tourists to choose rural tourism destinations. However, tourism experience memory is a multi-dimensional concept. At present, scholars have not reached a consensus on the theory and measurement of tourism experience memory [51]. The existing scale also fails to fully capture what factors enable tourists to generate rural tourism experience memory [9]. There are also large differences in research settings [2,52] and sampling design [49,53]. Although tourism experience memory is an important and growing area of research, research on this topic remains uncertain and fragmented [54,55]. In recent years, researchers have studied the factors influencing visitor memory formation through visitor tourism experiences in museums and historic Sites [56,57,58]. However, the current academic research on rural tourism experience memory lacks a cohesive and timely synthesis. The theoretical model of this study can explain the relationship between rural tourism experience memory and its antecedents. Through the induction of the necessary nodes of rural tourism experience memory into the framework, the structural relationship between rural tourism experience memory and its antecedents is clearly revealed.

First, this paper summarizes and fills two essential nodes in the antecedent relationship of rural tourism experience memory: supply perception and positive emotions, and proposes four aspects of supply perception, three types of emotions, and two kinds of memory characteristics applicable to rural tourism destination. After the evaluation and modification of several models by experts, a variable measurement scale that is in line with the characteristics of rural tourism destinations was designed. This is consistent with the research conclusions of Otto et al. who believed that the memory of tourist experience was based on tourists’ assessments of their trips [59].

Second, this paper further elaborates the different effects of rural tourism supply perception on various positive emotions, indicating that rural tourism supply perception is the power source of the formation of rural tourism experience memory, and reveals the mechanism between the two. The above research contents have verified the relevant results of existing studies to different degrees [32,33,34,36].

Third, this paper deduces and tests the structural relationship between positive emotions and rural tourism experience memory. According to the test results of Hypothesis 2, joy generated through rural tourism experience can form reproducible long-term memories, which is consistent with the fundamental starting point of tourists’ pursuit of joy. The above findings comprehensively verify the research results of Tung, Pan, She, Kim et al. and reveal the validity of three kinds of positive emotions in the formation process of rural tourism experience memory [9,12,13,37]. This means that the refined positive emotions not only reflect the diversity of the perceived driving effect of rural tourism supply, but also have the validity to form memories of the rural tourism experience.

Fourth, this paper deduces and tests the structural relationship between rural tourism supply perception and rural tourism experience memory. This verifies that not only can perception about rural tourism supply drive the generation of a variety of positive emotions, but it can also have a significant effect on the formation of rural tourism experience memories. This is consistent with the relevant research results of Lee, Pan, Barbieri, and others [13,23,39,60,61,62]. This again verifies that there are three kinds of coexisting structural relations in the formation process of rural tourism experience memories rather than a single linear relationship.

## 6. Conclusions

### 6.1. Findings

First, this paper deduces and tests the structural relationship between rural tourism supply perception and positive emotions. According to the research results of Hypothesis 1, perception about the rural tourist attraction has a driving effect on the generation of positive emotions, and the influence intensity is followed by joy, positive surprise, and love in this order. This indicates that tourists are more likely to have interactive emotions when they come into contact with other tourists, local residents, and tourism service personnel while using the tourist facilities and enjoying the services. The promotion perception has a driving effect on the generation of positive emotions, and its influence is through love and positive surprise, which indicates that tourists’ promotion perception includes perception about interactivity with tourism service personnel, obtainability of service, and advice from tourism practitioners. Further, they can better induce the interaction of tourists. Information perception has a driving effect on the generation of positive emotions, and its influence generates joy, love, and positive surprise, indicating that tourists perceive that they can obtain tourism information from multiple channels before going to the destination, which would force the tourists to imagine the destination, and their positive expectations would induce the generation of positive emotions.

Second, positive surprise generated in the process of rural tourism experience can make the long-term memories of tourists both vivid and reproducible. Tourists tend to look for freshness and excitement in rural tourism destinations to meet their inner needs for positive surprise, and positive surprise in the tourism experience is often the most beautiful and easily recalled memory for tourists. Rural tourism can provide visitors with long-term memories characterized by vitality and reproducibility, which is realized through communication and interaction with other tourists and local residents.

Third, according to the test results of Hypothesis 3, tourist attraction perception, promotion perception, and information perception have a significant impact on the viability of tourists’ long-term memories about their experience, while service/facility perception does not affect the viability of tourists’ memories. Tourist attraction perception, service/facility perception, and information perception also have significant influences on the reproducibility of tourists’ long-term memories about their experience, while promotion perception does not.

Fourth, positive emotions play an important mediating role in the influence of rural tourism supply perception on rural tourism experience memory. According to the test results of Hypothesis 4, regarding the influence of resources perception on rural tourism experience, positive surprise has the strongest intermediary effect. This suggests that the inclusion of natural landscapes, cultural landscapes, and rural tourist attraction, if consciously designed and offered as surprise elements, can maximize the novelty, shock, and a sense of positive surprise in tourists, effectively promoting the vividness of tourists’ memories. From the perspective of the effect of service/facility perception on rural tourism experience memory, love is the strongest mediator, indicating that the memories of tourists are mainly dominated by the interactive emotions brought on by local service and facility supply. From the perspective of the influence of information perception on rural tourism experience memory, positive surprise plays the strongest mediating role, which means that the information that can bring surprise to tourists is more valuable and results in the formation of unforgettable experience memories. From the perspective of the effect of promotion perception on rural tourism experience memory, the mediating effect of positive emotion only exists in the formation of tourists’ vivid experience memory, and positive surprise from promotion activities is more likely to form vivid long-term memories.

### 6.2. Management Implications

This study reveals the theoretical mechanism of the formation of rural tourism experience memory. This paper provides a more detailed description of tourists’ positive emotions. It also tests the effectiveness of multiple positive emotions through the vividness and reproducibility of rural tourism experience memories. Additionally, this paper reveals that rural tourism supply perception can directly generate rural tourism experience memories even without the positive emotion variable, which provides a practical direction. Based on the above research results and conclusions, this paper puts forward the following management recommendations:(1)Exploit the potential of rural tourist attraction. As a world cultural heritage site, Xidi Village has profound historical and cultural value. The combination of rural natural scenery and ancient residential buildings is rare worldwide and cannot be replicated. However, at present, tourism in Xidi Village is still based on sightseeing. Tourists focus on the green mountains, clear waters, white walls, and black tiles that characterize the destination. Only a few activities can immerse tourists, and there is a lack of emotionally engaging activities and memory points of on-site experience. Xidi Village managers should take advantage of tourist attractions, increase entertaining and interactive experience projects. They could add 3D projection effects to help visitors intuitively understand the site’s characteristics, history, and architectural construction methods. Brochures, handicrafts, as well as leisure postcards, can be used to help tourists connect with the local culture and form unique memories.(2)Managers should constantly improve the service level of local tourism service personnel. According to the previous research conclusion, it is better for managers to spend more labor, material, and financial resources on services and facilities than to pay more attention to the interaction with tourists at the current limited service and facilities level. However, the reverse would be more conducive to the formation of tourists’ rural tourism experience memories. High-quality service can bring more psychological satisfaction to tourists. A simple greeting from service staff, a thoughtful reservation, and timely and effective communication and interaction will enhance the positive emotions of tourists, enabling the tourists to fully enjoy both the natural ecological environment and cultural atmosphere, forming strong tourism experience memories.(3)Destination managers should pay attention to the role of information and promotion. Tourists’ information and promotion perceptions have an important effect on their pre-tour experiences. Managers should pay attention to conveying the potential positive feelings one may experience from visiting their destinations, and thus, encourage the formation of positive expected experiences among tourists. Managers can take the memory points of tourism consumers as promotional information, design powerful experiential marketing messages to stimulate tourists, and guide tourists to beautify and reconstruct their memories. Rural tourism destination managers should hire site management professionals to promote rural tourism characteristics, using apps such as WeChat, Weibo, BBS, and other new media to spread information about rural tourism products, develop new media marketing effects, improve related tourism products in the industry, and drive awareness.

### 6.3. Research Deficiencies and Prospects

Our study results showed that more than half of the respondents were students on academic excursions. Therefore, further research should focus on various types of tourists and the different characteristics of the respondents. This study only conducted a survey on tourists’ perceptions of destination supply shortly after the trip ended. Therefore, it is not clear whether the strong emotions felt during rural tours last; this requires further research.

## Figures and Tables

**Figure 1 behavsci-12-00475-f001:**
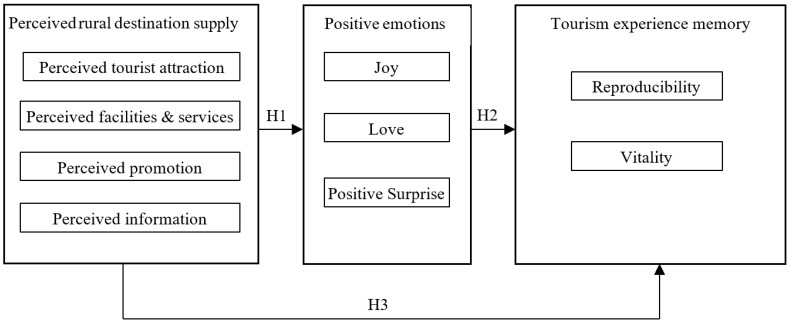
Theoretical model of the relationship between the antecedents of rural tourism experience memory.

**Figure 2 behavsci-12-00475-f002:**
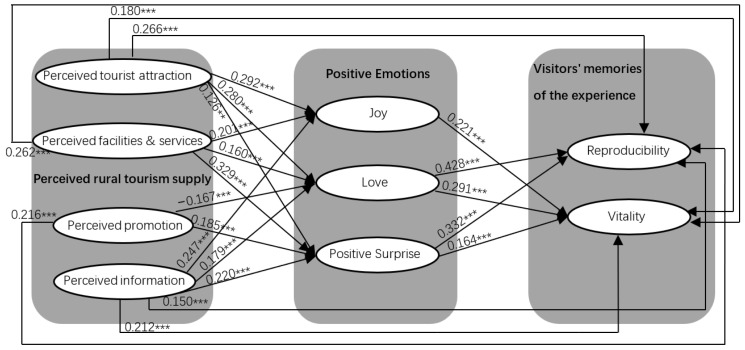
The action mechanism of antecedent variables of rural tourism experience memory. *** *p* < 0.001.

**Table 1 behavsci-12-00475-t001:** Demographic and travel characteristic of respondents (N = 556).

Characteristic	Frequency	%
Duration of stay		
1–1.5 days	377	67.9
1.5–2 days	133	23.8
Over 2 days	46	8.3
Transportation used		
Tourist Coach	368	66.1
Bus	51	9.1
Car	127	22.8
Other	10	2.0
Marital status		
Single	376	67.7
Married	180	32.3
Occupation		
Student	286	51.6
Education	87	15.5
Service industries	49	8.7
Business	29	5.3
Housekeeper	24	4.3
Freelancer	20	3.7
Others	11	2.0
Retired	50	8.9
Information about rural tourist areas		
Relatives and friends	315	56.5
Personal experience	73	13.2
Internet	117	21.1
Television/Magazine	40	7.1
Travel agencies	11	2.1

**Table 2 behavsci-12-00475-t002:** Reliability analysis results.

Nodes	Variables	Items	Standardized Load	Mean	Std.D	T-Value	α	AVE	CR
Destination Supply perception	tourist attraction perception	Natural Landscape	0.692	6.01	0.841	13.429	0.750	0.524	0.768
		Spatial Pattern	0.741	5.31	0.999				
		Culture and Art	0.738	5.44	1.207				
	Service/facility perception	Service Quality	0.662	4.77	1.194	19.399	0.741	0.536	0.821
		Service Attitude	0.664	4.94	1.170				
		Service Facilities	0.749	5.22	1.101				
		Communal Facilities	0.840	5.11	1.291				
	Information perception	Inquiry Channels	0.850	5.37	1.090	13.777	0.716	0.585	0.807
		Intelligent Tourism Information Services	0.698	5.24	1.150				
		Intensive Advertising	0.738	4.72	1.252				
	Promotion perception	Promotional Activity	0.847	4.55	1.290	13.441	0.745	0.641	0.780
		Promotion Scheme	0.751	4.92	1.163				
Positive emotions	Joy	Carefree	0.806	5.71	1.145	11.827	0.755	0.614	0.826
		Delight	0.837	5.25	1.111				
		Fun	0.701	5.11	1.167				
	Love	Friendliness	0.752	5.16	1.194	23.411	0.776	0.572	0.842
		Enthusiasm	0.798	4.94	1.170				
		Gratitude	0.757	5.22	1.101				
		Feel at Ease	0.716	5.11	1.291				
	Positive surprise	Novelty	0.696	5.12	1.145	25.616	0.807	0.522	0.813
		Shock	0.772	4.55	1.320				
		Revel	0.670	5.09	1.215				
		Meaning	0.746	5.38	1.176				
Tourism experience memory	Reproducibility	Reproducing the Tourist Experience Recall	0.761	5.55	0.969	36.755	0.753	0.541	0.825
		Participating in the Activity Recall	0.735	5.38	1.041				
		Emotional Feeling Recall	0.748	5.36	1.092				
		Major Scenic Spots Recall	0.699	5.48	1.037				
	Vividness	Overall Process	0.861	4.84	1.163	25.805	0.799	0.630	0.872
		The Layout and Structure of the Main Scenic Spots	0.766	5.07	1.118				
		The Sounds They Heard	0.782	4.57	1.275				
		The Scenes They Saw	0.762	5.18	1.203				

**Table 3 behavsci-12-00475-t003:** Differential validity among perceived variables of rural tourism destination supply.

	Tourist Attraction Perception	Service/Facility Perception	Information Perception	Promotion Perception
Tourist attraction perception	0.724			
Service/facility perception	0.411 **	0.732		
Information perception	0.365 **	0.455 **	0.764	
Promotion perception	0.257 **	0.385 **	0.383 **	0.800

** *p* < 0.01.

**Table 4 behavsci-12-00475-t004:** Differential validity among positive emotion variables.

	Joy	Love	Positive Surprise
Joy	0.783		
Love	0.525 **	0.756	
Positive surprise	0.637 **	0.448 **	0.722

** *p* < 0.01.

**Table 5 behavsci-12-00475-t005:** Differential validity among variables of tourism experience memory.

	Reproducibility	Vividness
Reproducibility	0.736	
Vividness	0.497 **	0.793

** *p* < 0.01.

**Table 6 behavsci-12-00475-t006:** Mediating effect analysis of research paths.

Logical Path	Mediating Effect	Total Effect	Mediating Effect/Total Effect
Rural tourist attraction perception → Joy → Reproducibility	0.047	0.286	0.164
Rural tourist attraction perception → Love → Reproducibility	0.017	0.286	0.059
Rural tourist attraction perception → Love → Vividness	0.023	0.180	0.128
Rural tourist attraction perception → positive surprise → Reproducibility	0.073	0.286	0.255
Rural tourist attraction perception → Positive surprise → Vividness	0.098	0.180	0.544
Service/facility perception → Joy → Reproducibility	0.032	0.122	0.262
Service/facility perception → Love → Reproducibility	0.045	0.122	0.369
Service/facility perception → Love → Vividness	0.061	0.262	0.233
Service/facility perception → Positive surprise → Reproducibility	0.042	0.122	0.344
Service/facility perception → Positive surprise → Vividness	0.056	0.262	0.214
Information perception → Joy → Reproducibility	0.040	0.212	0.189
Information perception → Love → Reproducibility	0.030	0.212	0.142
Information perception → Love → Vividness	0.040	0.150	0.267
Information perception → Positive surprise → Reproducibility	0.047	0.212	0.221
Information perception → Positive surprise → Vividness	0.063	0.150	0.420
Promotion perception → Love → Vividness	0.034	0.216	0.157
Promotion perception → Positive surprise → Vividness	0.058	0.216	0.268

## Data Availability

The data analyzed in this paper are proprietary, and therefore cannot be posted online.

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
