# Peer review of "Antecedents of Rural Tourism Experience Memory: Tourists’ Perceptions of Tourism Supply and Positive Emotions"

_behavsci, 2022, doi:10.3390/bs12120475_

Round 1

Reviewer 1 Report

This is an interesting study on rural tourism experience memory.

The empirical background sheds some light on a relatively new dimension in tourism studies. However, a major revision is required, especially in terms of better grounding this paper in the existing literature. For instance, authors state that  'there is rare research focusing on the exploration of the factors that affect tourism experience memory', but in fact there are several such studies which have been done exactly on this topic and some of them were focused on focus groups with students too (see the studies of Zalut L., 2018 - doi - 10.1080/10598650.2017.1419412,Light D. el al. (doi - 10.1080/14683857.2019.1702619, doi- 10.1080/15387216.2019.1581632, even one published in journal Societies which showed the role of museum and transitional justice via students experience and memory - https://www.mdpi.com/2075-4698/11/2/43/htm). Even domestic tourism connected to transitional justice for certain tourist sites (eg museums, heritage buildings and so on) could be important in relation to memory of visitors (see a paper in Current Issues of Tourism, 2021). So, a more engagement with sources like the above ones have to be presented also in the discussions.

The empirical part of the paper is very good, even the conclusions are strong. Discussions should stand as a separate section because now the section is entitled Conclusion and discussion.

The reference list should be enlarged because there are only 50 references in the reference list.

Reviewer 2 Report

The manuscript is well written, it is relatively concise in the description, but a little confusing in the methodological part and in the explanation of the hypotheses.

The manuscript as it is presented lacks a concise objective at the end of the introduction. The last sentence of the first paragraph of the introduction should move to the end of the introduction.

In the second section, only the literature review should be included as a subtitle. Further on, after reference 18, what is written there is not a literature review. This part should be allocated either in the methodological part, or if it is repeated, it should simply be excluded from the manuscript.

In the methodological part, what follows the instrumentation seems to be an orphan phrase. Here, more information should be added, or the phrase must be deleted.

Further on, the “7-level Likert scale” is referred to several times. This is not the most correct term and should be changed to “7-point Likert scale”. Even further on, the word “poiny” appears, which must have been a small typo in relation to the word “point” and which should also be amended.

In the sub-section “positive emotions” appears the word “intoxication” which apparently seems to be an inappropriately used word. Please amend.

Before the “The sample” section there seems to be a lack of words a sentence above. Please check, if not missing, close the sentence with a full stop “.”.

Table 1 is very confusing. I suggest that it should be divided into two or more smaller tables, or alternatively use a histogram for example for the age distribution of the sample.

The “Conclusion and discussion” section is not correct. They must be separate, and the discussion part must appear first. The conclusion should appear with the “Management implications”.

At the end of page 14 appears the word “advertising” which should be changed to the word “advertising”.

Some references are not correctly introduced at the end. Examples are refs 40, 41, 42 and possibly some more. Please verify.

Round 2

Reviewer 1 Report

Authors improved the content of this paper, so I am happy to accept the paper for publication.

Just for the production stage of the paper authors have to attentively check all the article references from the reference list in order to have volume, issue, page numbers and doi numbers. 

Reviewer 2 Report

The manuscript (MS) has improved considerably after the suggested changes. The inclusion of more literature where it was scarce made the MS much stronger. The correction of the few typos that were detected throughout the body of text demonstrates that care was taken to improve the text and its readability.

The old Table 1 was removed and the information was included in the body of the text. I believe the table just needed to be improved to make it easier to read or alternatively replaced with more explanatory figures/histograms.
